# Identification of differentially expressed genes involved in amino acid and lipid accumulation of winter turnip rape (*Brassica rapa* L.) in response to cold stress

Yan Fang[1,2], Jeffrey A. Coulter[3], Junyan Wu[1,2], Lijun Liu[1,2], Xuecai Li[1,2], Yun Dong[4], Li Ma[1,2], Yuanyuan Pu[1,2], Bolin Sun[1,2], Zaoxia Niu[1,2], Jiaojiao Jin[1,2], Yuhong Zhao[1,2], Wenbo Mi[1,2], Yaozhao Xu[5], Wancang Sun[1,2]*

**1** Gansu Provincial Key Laboratory of Aridland Crop Science, Lanzhou, China, **2** College of Agronomy, Gansu Agricultural University, Lanzhou, China, **3** Department of Agronomy and Plant Genetics, University of Minnesota, St. Paul, MN, United States of America, **4** Crop Research Institute, Gansu Academy of Agricultural Sciences, Lanzhou, China, **5** College of Agronomy and Biotechnology, Hexi University, Zhangye, China

* 18293121851@163.com

**Data Availability Statement:** All relevant data are within the paper and its Supporting Information files.

## Abstract

Winter turnip rape *(Brassica rapa* L.) is an important overwintering oil crop that is widely planted in northwestern China. It considered to be a good genetic resource for cold-tolerant research because its roots can survive harsh winter conditions. Here, we performed comparative transcriptomics analysis of the roots of two winter turnip rape varieties, Longyou7 (L7, strong cold tolerance) and Tianyou2 (T2, low cold tolerance), under normal condition (CK) and cold stress (CT) condition. A total of 8,366 differentially expressed genes (DEGs) were detected between the two L7 root groups (L7CK_VS_L7CT), and 8,106 DEGs were detected for T2CK_VS_T2CT. Among the DEGs, two ω-3 fatty acid desaturase (*FAD3*), two delta-9 acyl-lipid desaturase 2 (*ADS2*), one diacylglycerol kinase (*DGK*), and one 3-ketoacyl-CoA synthase 2 (*KCS2*) were differentially expressed in the two varieties and identified to be related to fatty acid synthesis. Four glutamine synthetase cytosolic isozymes (*GLN*), serine acetyltransferase 1 (*SAT1*), and serine acetyltransferase 3 (*SAT3*) were down-regulated under cold stress, while S-adenosylmethionine decarboxylase proenzyme 1 (*AMD1*) had an up-regulation tendency in response to cold stress in the two samples. Moreover, the delta-1-pyrroline-5-carboxylate synthase (*P5CS*), δ-ornithine aminotransferase (δ-*OAT*), alanine-glyoxylate transaminase (*AGXT*), branched-chain-amino-acid transaminase (*ilvE*), alpha-aminoadipic semialdehyde synthase (*AASS*), Tyrosine aminotransferase (*TAT*) and arginine decarboxylase related to amino acid metabolism were identified in two cultivars variously expressed under cold stress. The above DEGs related to amino acid metabolism were suspected to the reason for amino acids content change. The RNA-seq data were validated by real-time quantitative RT-PCR of 19 randomly selected genes. The findings of our study provide the gene expression profile between two varieties of winter turnip rape, which lay the foundation for a deeper understanding of the highly complex regulatory mechanisms in plants during cold treatment.

**Funding:** This work was supported by Gansu Provincial Key Laboratory of Aridland Crop Science, Gansu Agricultural University (No. GSCS-2017-10); the National Natural Science Foundation of China (No. 31860388); the Utilization Technology of Rapeseed Heterosis and Creation of Strong Heterosis of China (No. 2016YFD0101300) and the Agriculture Research System of China (CARS-12).

**Competing interests:** The authors have declared that no competing interests exist.

## Introduction

Plants must cope with various unfavorable growth conditions such as cold, drought, and salinity, which adversely influence their growth development, yield, and quality [1]. Cold is one of the major adverse environmental stresses facing many crops. To survive under cold conditions, plants have developed specialized survival strategies. Generally, plants can acquire increased freezing tolerance when exposed to low temperatures for an extended period of time [2]. This highly complex process is associated with morphological, molecular, biochemical, and physiological changes [3]. which eventually exhibit significant changes in gene expression levels and metabolites [4, 5].

Transcriptome analysis is now an effective method for cold stress investigation in many plants, and studies in rice [6], *Lotus japonicus* [7], *Anthurium* [8], and *Brassica oleracea* [9] identified a number of genes that responded to cold stress. Many genes related with sugar metabolism, antioxidant defense system, plant hormone signal pathway, transcription factor, and photosynthesis have been reported to respond to cold stress in *Populus tomentosa* [10]. In addition, changes in the expression level of some genes related with lipid synthesis and amino acid synthesis play an important role in plant response to cold stress [11, 12]. Increased expression of *OsSPX1* was reported to enhance cold tolerance in tobacco (*Nicotiana tabacum* L. cv. Xanthinn) and *Arabidopsis thaliana* [13]. The *AP2/EREBP* transcription factor, referred to as OsDREB, acts on response to cold stress in rice [14]. Another *AP2/ERE* family member, C-repeat/dehydration responsive element-binding factors (CBPs) in *Arabidopsis*, were also found to play prominent roles in responding to low temperature [15, 16]. The accumulation of the precursor geranylgeranyl chlorophyll *a* and down-regulation of *GERANYLGERANYL REDUCTASE* (*GGR*) were found to help maize cope with chilling stress [17]. Recent studies also reveal that various signaling pathways were activated in response to cold stress; for example, genes involved in gibberellin, ethylene, auxin, and the flavonoid signal pathway can be affected by low temperature [18–21]. Despite many genes having been identified as responsive to cold stress, researchers have been unable to decipher the cold stress response mechanism. Thus, more mechanisms of plant cold resistance need to be explored.

Winter turnip rape (*Brassica rapa* L.) is a cruciferous oilseed crop that can adapt to cool climates due to its strong root system [22]. The sowing time of winter turnip rape in cold and arid regions of northwestern China is in the middle of August. At the four to five true leaf stage, the plants experience a reduction in air temperature with the onset of autumn, and at the eight to nine true leaf stage the aboveground parts of winter turnip rape start to wither, while the roots begin to overwinter. In the following spring, winter turnip rape begins to turn green. Thus, cold tolerance of the roots is very important for winter turnip rape to survive in winter [23]. Our previous work revealed important physiological changes of winter turnip rape cultivars in response to cold stress [24], but the molecular mechanisms underlying cold stress response in winter turnip rape are still unclear.

In order to reveal the molecular mechanism by which winter turnip rape copes with cold stress and to discover genes that may be useful in breeding for cold-tolerant varieties, the roots of two winter turnip rape varieties, Longyou7 (L7) and Tianyou2 (T2), were used as materials to perform transcriptome analyses. This study provides insight into differentially expressed genes (DEGs) involved in lipid and amino acids accumulation in winter turnip rape in response to cold stress, and serves as a theoretical basis for the genetic improvement of cold-tolerant varieties that can be used as a reference for many other crops.

## Materials and methods

### Plant materials and treatments

Two cultivars of winter turnip rape, Longyou7 (L7) and Tianyou2 (T2), were used in this study. Longyou7 is an ultra-cold-tolerant cultivar, with an over wintering rate above 80% in

cold and arid regions of northwestern China [25]. Tianyou2 is a weak cold-tolerant cultivar, with an over-wintering rate ranging from 40% to 80%. Seeds were first germinated and cultivated in plastic pots (9.0 cm deep and 7.5 cm in diameter) filled with matrix (1:1 ratio of brown coal soil: sand) under typical conditions in Lanzhou City (36˚ 7' N, 103˚7' W), Gansu Province, China, 60 days after sowing, then the seedlings were treated with different conditions including control (CK) and cold stress. In the CK conditions, seedlings were grown in a growth chamber under 22˚C (12 h light/12 h dark) for 48 h. For the cold stress treatment (CT), plants were transferred to a low temperature light incubator with intensity of illumination 6000 Lx and 60% relative humidity and the temperature in the incubator was gradually decreased by 2˚C h-1 from 22˚C to -4˚C. When the temperature reached -4˚C, the plants were kept at -4˚C for 6 h and an obvious phenotypic difference was observed between L7 and T2. After treatments, the roots were carefully cut with a razor blade, collected and frozen immediately in liquid nitrogen, and stored at -80˚C for further use. Each sample consisted of roots from six plants grown in the same condition, and there were three biological replicates for transcriptomics analysis.

## Determination of free fatty acid (FFA) and amino acid

**Determination of free fatty acid.** The plant FFA ELISA Kit (ADS-W-ZF001, Jiangsu Kete Biotechnology Co. Beijing, China) was used to determine the content of free fatty acid as previously described [26]. A plant sample of 50 mg was ground with sodium sulfate (anhydrous), then the surface moisture was absorbed with absorbent paper according to the tissue quality. The volume of the extraction solution (mL) was 1:5 ~ 10 to homogenize in an ice bath. Samples were then centrifuged at 8000 rpm at 4˚C for 10 min, then the supernatant was collected and tested. The plant FFA ELISA kit was used to determine fatty acid content, dilute method of original density standard was shown in S2 Table. Blank wells were set separately (blank comparison wells do not add sample and HRP-Conjugate reagent, other each step operation is same). Next, the standard, sample dilution, and testing sample were added orderly and incubated for 30 min at 37˚C. Then, HRP-Conjugate reagent was added, the samples were incubated for 30 min at 37˚C, and washed five times. Next, Chromogen Solution A and B was added, the samples were incubated for 10 min at 37˚C, and washed five times. Finally, stop solution was added and samples were read at an absorbance of 450 nm within 15 min. Standard density and optical density (OD) values were used to develop the standard curve and obtain a linear regression equation that was used for calculating the actual sample density. Three biological replicates were included for FFA analysis.

**Determination of amino acid.** We selected 17 amino acids including aspartic acid, glutamate, serine, glycine, histidine, arginine, threonine, alanine, proline, tyrosine, valine, methionine, cystine, isoleucine, leucine, phenylalanine, and lysine. The standards of these 17 amino acids were obtained from Shanghai Ambrosia Pharmaceutical Co. For amino acid determination, the roots of each sample were dried in an oven at 80˚C and then pulverized. Next, 0.13 g of sample and 8 ml of 0.02 mol $L^{-1}$ of hydrochloric acid were placed into a 10 mL volumetric flask, vortexed for 5 min, ultrasonically extracted for 10 min, then well mixed with making up the volume to 10 ml and kept in the dark for 2 h. Then, 5 mL of the solution was centrifuged at 4000 rpm for 10 min and 1 mL of the supernatant was removed. Next, 1 mL of 6–8% sulfosalicylic acid, 250 μL of 1 mol $L^{-1}$ triethylamine acetonitrile solution, and 250 μL of 0.1 mol $L^{-1}$ phenyl isothiocyanate acetonitrile solution were added to the 1 mL of supernatant in sequence and mixed at room temperature for 1 h. Then, 2 mL of n-hexane was added, followed by violent shaking, and the sample was left undisturbed for 10 min. Finally, the solution was filtered through a 0.22-μm micro pore membrane for analysis.

A Universal C18 (Shiseido, Tokyyo, Japan.) chromatographic column (250 mm × 4.6 mm 5 μm) was used for determination of amino acid content. For the mobile phase, A was sodium acetate solution (3% acetonitrile and 97% of 0.1 mol L$^{-1}$ anhydrous sodium acetate adjusted to a pH of 6.5) and B was acetonitrile and water with 4:1 volume ratio. The mobile  phase gradient  elution  procedure is shown in S3 Table, which was delivered at 1.0 mL min$^{-1}$. The injection volume was 10 μL. Three biological replicates were included for amino acid analysis.

## RNA extraction and library construction for illumina sequencing

Total RNA of the two cultivars in the two treatments was extracted using an RNAprep Pure Plant kit (Bio TeKe, Beijing, China) following the manufacturer's instructions with three biological replicates. RNA concentration was measured using a NanoDrop 2000 (Thermo). RNA integrity was assessed using the RNA Nano 6000 Assay Kit of the Agilent Bioanalyzer 2100 system (Agilent Technologies, CA, USA). A total amount of 1 μg RNA per sample was used as input material for RNA sample preparation. Sequencing libraries were generated using NEB Next Ultra$^{TM}$ RNA Library Prep Kit for Illumina (NEB, USA) following manufacturer's recommendations and index codes were added to attribute sequences to each sample. PCR products were purified (AMPure XP system) and library quality was assessed on the Agilent Bioanalyzer 2100 system. The clustering of the index-coded samples was performed on a cBot Cluster Generation System using the TruSeq PE Cluster Kit v4-cBot-HS (Illumina) according to the manufacturer's instructions. After cluster generation, the library preparations were sequenced on an Illumina platform and paired-end reads were generated. Pearson's rank correlation analysis was performed to evaluate the reproducibility among biological replicates.

## De novo assembly of the transcriptome

Raw data (raw reads) of fastq format were firstly processed through in-house per l scripts. In this step, clean data (clean reads) were obtained by removing reads containing adapter, ploy-N, and low quality from raw data. At the same time, Q30, GC content, and sequence duplication level of the clean data were calculated. All downstream analyses were based on clean data with high quality. The clean reads were aligned to the reference genome sequences of the *Brassica rapa* genome (brassicadb.org/Bra_Chromosome_V1.5/) using HISAT2 software with default parameters [27]. Only reads with a perfect match or one mismatch were further analyzed and annotated based on the reference genome. A reference-based assembly of all reads was performed using String Tie, which produced a set of transcripts as small as possible.

## Expression level analysis and gene annotation

Using BLAST [28], sequence alignment was conducted between the identified new genes and databases of NR [29], Swiss-Prot [30], GO [31], COG [32], KOG [33], Pfam [34], and KEGG [35]. GO, KEGG, and NR were our most important referenced databases for further research.

Gene Ontology (GO) enrichment analysis of the DEGs was implemented by the GO seq R packages based on the Wallenius non-central hyper-geometric distribution, which can adjust for gene length bias in DEGs. KEGG is a database resource for understanding high-level functions and utilities of the biological system, such as the cell, organism and ecosystem, from molecular-level information, especially large-scale molecular datasets generated by genome sequencing and other high-throughput experimental technologies (http://www.genome.jp/kegg/). We used KOBAS [36] software to test the statistical enrichment of DEGs in KEGG pathways. Fragments per kilobase of transcript per million fragments mapped (FPKM) was used as an indicator of transcription or gene expression [37]. The criteria for detecting DEGs was |log2(fold change)| ≥1 and false discovery rate (FDR) <0.01.

## Quantitative Real-Time PCR (qRT-PCR)

Genes of interest were selected for further confirmation by qRT-PCR analysis following the manufacturer's protocols (SYBR Green I; Lumiprobe). Total RNA was extracted and purified as above. Gene specific qRT-PCR primers were designed using Primer-BLAST (http://blast.ncbi.nlm.nih.gov/Blast.cgi), and the sequences are listed in S4 Table. All quantifications were normalized to Actin-7 (Accession no. XM_009127096.2) and calculated using the $2^{-\Delta\Delta Ct}$ method [38]. Three biological replicates were used for qRT-PCR analysis.

# Results

## Phenotypic and physiological responses of the two varieties to cold stress

The variation of phenotypic traits and physiological responses were used for evaluated the cold resistance in L7 and T2. Previously, we had evaluated the over-wintering rate, which is an indicator for cold tolerance of L7 and T2 in field trials in different environments [39]. Significant differences in over-wintering rate were detected between L7 and T2, with an over-wintering rate >81% observed for L7, while an over-wintering rate ranging from 28.50% to 87.90% for T2 (S1 Table). Morphology and physiological characteristics research showed that dry matter accumulation in L7 was higher than T2 during cold acclimation [23], especially the root system of L7 is larger than T2 before winter (Fig 1A). Under low temperature condition, the leaf tissues of winter rape must be damaged firstly. No morphological differences were detected between L7 and T2 at 22˚C. However, an obvious phenotypic difference was observed between L7 and T2 after cold treatment (-4˚C), as cultivar L7 remained green and grew normally, while the leaves of T2 were injured and exhibited various symptoms of freezing injury, including leaf margin wilting, drying of leaf edges, and small white spots. This result indicates that L7 has greater cold tolerance than T2 (Fig 1B).

Free fatty acid (FFA) and seventeen amino acids contents were used as physiological indicators to evaluate cold resistance. Under cold stress, FFA content was increased in the two varieties, and the same trend was found for the content of aspartic acid, glutamate, serine, histidine, and alanine (Fig 2). Among these, glutamate, serine, and alanine were significantly increased after cold treatment. However, only three amino acids including threonine, methionine, and leucine decreased with cold treatment in the two varieties. The other nine amino acids, including glycine, arginine, proline, tyrosine, valine, cystine, isoleucine, phenylalanine, and lysine, were increased in L7 and decreased in T2 after cold stress (Fig 2).

## RNA-seq of winter turnip rape transcriptome

We used the Illumina Hiseq2500 platform for transcriptome sequencing. A total of 97.85 Gb of clean data were obtained. The quality of base was mostly above Q30, with >85% of the reads having a quality score above Q30. The GC content (>47%) was nearly the same in all samples. StringTie was used for reads mapping [40], and the mapped rate ranged from 53.54% to 78.84% (Table 1).

To assess the randomness of sequencing as well as find DEGs, three biological replicates were performed in our study. Meanwhile, Pearson's correlation coefficient was used to evaluate the reproducibility of the biological replicates. The correlations within the samples of the three biological replicates ranged from 0.915 to 1 (S1 Fig). However, the correlations between inter-samples changed significantly, with a value ranging from 0.516 to 0.978. The closer the correlation coefficient is to 1, the stronger the correlation between two samples.

## Validation of qRT-PCR

Another method, qRT-PCR analysis, was also used to validate the quality of the RNA-seq data. The *Actin-7* (Accession no. XM_009127096.2) was selected as the internal control for

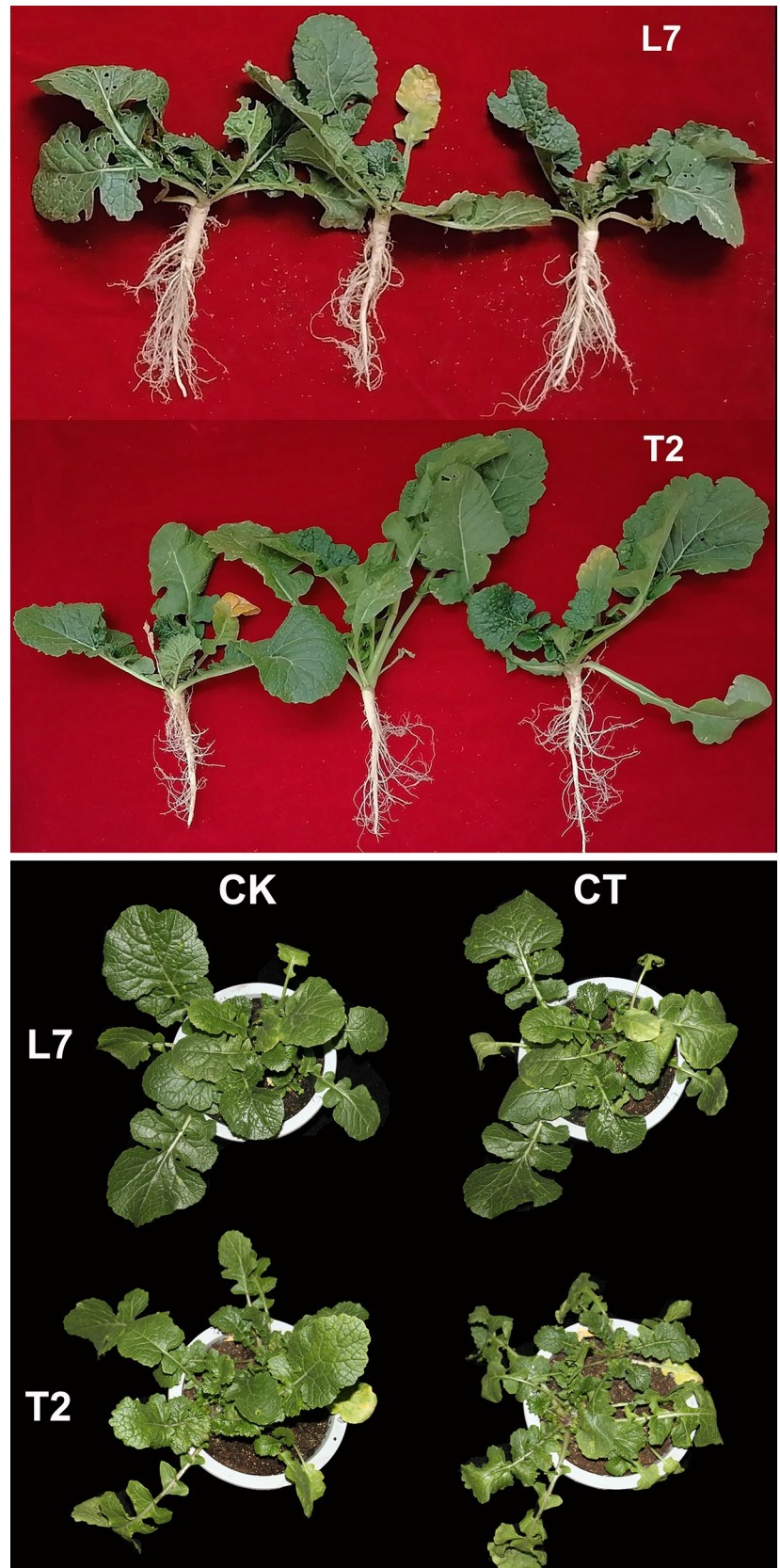

**Fig 1. Morphology and physiological characteristics of L7 and T2 under cold treatment.** (A) the root morphology of L7 and T2 cultivars. (B) the seedling morphology of L7 and T2 cultivars. CK refers to the control group of the sample at 22°C; CT refers to the cold treatment group of the sample at -4°C.

qRT-PCR validation. Nineteen genes were selected for qRT-PCR validation to verify the RNA-seq data, involving genes in the amino acid, sugar, lipid, and plant hormone signal metabolic pathways which have already been reported to be related with cold stress (S5 Table) [41–44].

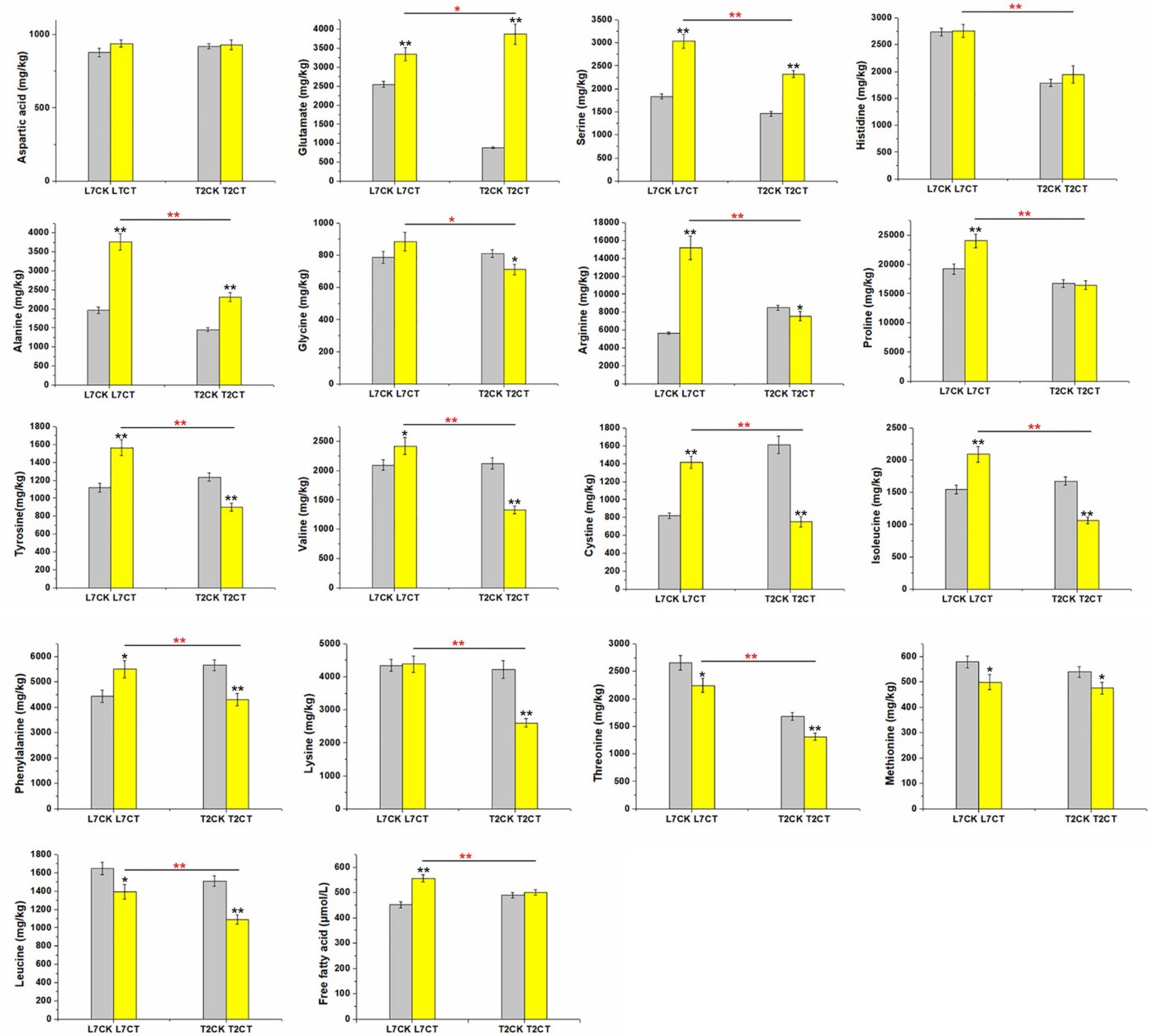

**Fig 2. Plant physiological indicators of free fatty acid and seventeen kinds of amino acids.** Error bars represent standard error of the mean. Significant difference between the control and treatment sample in the same variety at $P \leq 0.05$ and $P \leq 0.01$ are denoted by one and two black asterisks, respectively. Significant differences between two varieties under cold treatment at $P \leq 0.05$ and $P \leq 0.01$ are denoted by one and two red asterisks, respectively. L7CK refers to the control group of L7 at 22°C; L7CT refers to the cold treatment group of L7 at -4°C; T2CK refers to the control group of T2 at 22°C; T2CT refers to the cold treatment group of T2 at -4°C.

**Table 1. Overview of the winter turnip rape transcriptome data of L7 and T2 under control and cold treatment.**

| Transcriptome samples | Total clean reads | Clean bases (bp) | GC content | Q30 | Mapped reads |
|---|---|---|---|---|---|
| **L7CK-1** | 68,037,256 | 10,173,413,108 | 47.22% | 85.83% | 39,100,483 (57.47%) |
| **L7CK-2** | 41,784,302 | 6,250,028,276 | 47.40% | 85.03% | 32,708,276 (78.28%) |
| **L7CK-3** | 69,282,486 | 10,355,196,702 | 47.04% | 88.26% | 42,459,485 (61.28%) |
| **L7CT-1** | 50,006,420 | 7,477,648,424 | 47.31% | 85.01% | 39,424,872 (78.84%) |
| **L7CT-2** | 46,973,764 | 7,019,889,002 | 47.48% | 85.74% | 35,392,615 (75.35%) |
| **L7CT-3** | 69,983,050 | 10,464,320,412 | 47.60% | 87.69% | 50,110,506 (71.60%) |
| **T2CK-1** | 43,068,484 | 6,440,225,242 | 47.68% | 85.16% | 33,552,649 (77.91%) |
| **T2CK-2** | 42,338,952 | 6,328,571,160 | 47.49% | 85.01% | 32,884,928 (77.67%) |
| **T2CK-3** | 44,083,852 | 6,590,601,886 | 47.56% | 85.00% | 34,346,877 (77.91%) |
| **T2CT-1** | 55,172,864 | 8,245,631,848 | 47.23% | 86.29% | 34,880,629 (63.22%) |
| **T2CT-2** | 51,070,170 | 7,633,397,558 | 47.40% | 85.91% | 34,590,245 (67.73%) |
| **T2CT-3** | 72,712,420 | 10,871,525,802 | 47.10% | 87.44% | 38,931,922 (53.54%) |

Note: L7CK refers to the control group of L7 at 22°C; L7CT refers to the cold treatment group of L7 at -4°C; T2CK refers to the control group of T2 at 22°C; T2CT refers to the cold treatment group of T2 at -4°C; 1, 2, and 3 refers to three biological replicates, the same as in the following figures and tables.

The results of the RNA-seq and qRT-PCR analyses showed the same expression trends and were largely consist ($r = 0.9248$, $P < 0.05$) (S2 Fig), which indicate that the gene expression levels and DEGs obtained from transcriptome analysis were accurate and reliable.

## Analysis of expression level and identification of DEGs of winter turnip rape under cold stress

The expression level (FPKM) of transcripts in each sample was calculated and is shown in S1 Dataset. In total, we detected 8,366 (3,777 up-regulated and 4,589 down-regulated) and 8,106 (3,830 up-regulated and 4,276 down-regulated) DEGs in L7CK versus L7CT and T2CK versus T2CT. A total of 5,630 DEGs were common in both cultivars and were putatively considered to be associated with the phenotypic trait differences in this species (Fig 3A). The heatmap of all DEGs is shown in S3 Fig. The results show a large difference in gene expression pattern between L7 and T2 when they suffered from cold stress.

## KEGG enrichment analysis of DEGs

To investigate the mechanisms of DEGs in response to cold stress, we conducted KEGG annotation of these transcripts. We analyzed those DEGs that were specifically identified in L7 and T2. The KEGG enrichment analysis showed that L7 and T2 had different response pathways under cold stress. For L7, the up-regulated specific DEGs (S4A Fig) predominantly enriched in pathways of "RNA transport", "ribosome", "linoleic acid metabolism", "sphingolipid metabolism", "ether lipid metabolism", and "alpha-linolenic acid metabolism", and the down-regulated specific DEGs (S4B Fig) were highly enriched in "plant hormone signal transduction", "pentose and glucuronate interconversions", and "brassinosteroid biosynthesis" pathways. For T2, the up-regulated specific DEGs (S4C Fig) were highly enriched in "ribosome", "monoterpenoid biosynthesis", "phosphatidylinositol signaling system" (PI), and "Flavonoid biosynthesis", and the specific down-regulated DEGs (S4D Fig) were highly enriched in "phenylpropanoid biosynthesis", "glycerolipid metabolism", and "glycosphingolipid biosynthesis-ganglio series" pathways. Interestingly, the "sphingolipid metabolism", "other glycan degradation", and "sulfur relay system" pathways that were up-regulated in L7 after cold treatment were found to be down-regulated after cold treatment in T2. Previous research reported that

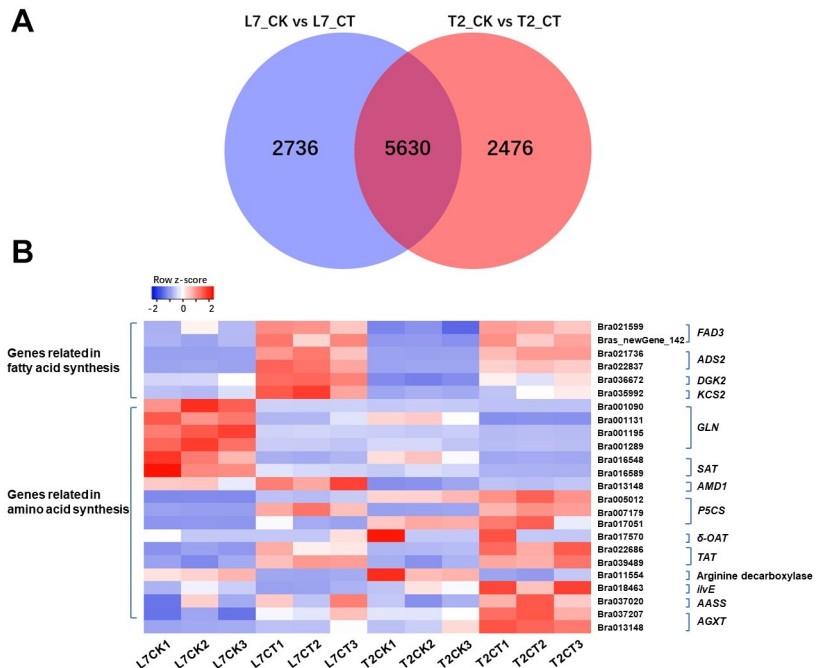

**Fig 3.** A. Venn graph of the number of differentially expressed genes in two winter turnip rape varieties. B. The expression pattern of the key genes involved in fatty acid synthesis and amino acids metabolism.

freezing tolerance in plants requires lipid remodeling to protect membranes [45], and that sugars and several amino acids prevent cellular dehydration [46], these may be the major reasons why L7 showed higher cold tolerance than T2.

## DEGs related to fatty acid synthesis

Plant lipids as a huge group exhibit great structural diversity from simple lipids to complex lipids and have a wide range of physical and chemical properties that are involved in a wide variety of physiological processes [47]. Lipid fatty acids play an important role in cold adaption [48], such as ω-3 fatty acid desaturase (*FAD3*), which was found to be associated with cold tolerance [49]. In the present study, we compared the fatty acid content between two varieties and identified several DEGs related to fatty acid synthesis. As a result, two *FAD3s*, Bra021559 and Brassica rape new gene142, were up-regulated under cold stress and more highly expressed in L7 (Fig 3B, S2 Dataset). The expression pattern of two delta-9 acyl-lipid desaturase 2 (*ADS2*) genes, Bra021736 and Bra022837, were found to be consistent with the above two genes and have been reported to be required for chilling and freezing tolerance in *Arabidopsis* [50]. These four desaturases all participate in biosynthesis of unsaturated fatty acids (ko01040) and fatty acid metabolism (ko01212) (Fig 4).

Phosphatidic acid is another important signaling lipid in cold stress responses in plants. Its accumulation could be conducted via a pathway of phosphorylation of diacylglycerol (DAG) by diacylglycerol kinase (*DGK*) [51]. Previous study has also shown DGK2 transcription to be induced in *Arabidopsis* in response to cold stress [52]. In this study, a *DGK2* (Bra036672) gene was up-regulated when the two varieties of winter turnip rape suffered from cold treatment (Fig 4, S2 Dataset). The expression level of this gene was higher in L7 compared to T2 during cold stress (Fig 3B). *DGK2* (Bra036672) was identified to participate in glycerolipid

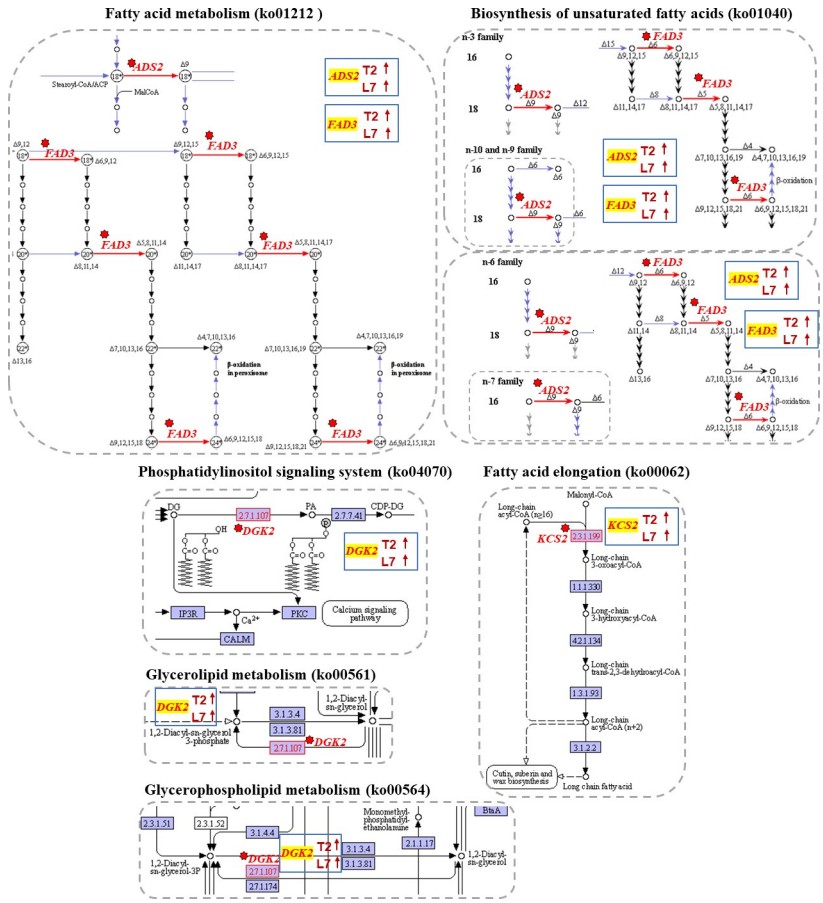

**Fig 4. KEGG pathway of DEGs related in fatty acid synthesis.** The images showed that the genes involved in fatty acid synthesis were mapped to the reference pathway using KEGG. The genes annotated in KEGG map were highlighted with red texts and red arrows. The blue box in each pathway showed the expression tendency of genes in two cultivars. The red on the blue box shows upregulated, the green indicates downregulated.

metabolism (ko00561), gycerophospholipid metabolism (ko00564), and the phosphatidylinositol signaling system (ko04070) (Fig 4).

The FPKM of a DEG encoding 3-ketoacyl-CoA synthase 2 (*KCS2*) (Bra035992) was detected to significantly increase in L7 compared to T2 under cold stress (Fig 3B, S2 Dataset). It has been reported that *KCS2* mediates the synthesis of very-long-chain fatty acids (22 to 26 carbons in length) and is involved in the biosynthesis of aliphatic suberin in roots [53]. *KCS2* was annotated in Fatty acid elongation (ko00062), and may be a functional gene in response to cold stress in this study (Fig 4).

## DEGs related to amino acid metabolism

The increasing of some amino acids, such as proline, aspartic acid, asparagine, glycine, and valine has been investigated in wheat seedlings in response to cold stress (4°C) [54]. The changes of amino acids content were triggered by some genes that catalyze a series of physiological and biochemical reactions. Glutamine synthetase cytosolic isozyme (*GLN*), as a catalyst, participated in the L-glutamate to L- glutamine reaction. Four DEGs (Bra001090, Bra001131, Bra001195, and Bra001289) that encode the *GLN* gene (EC:6.3.1.2) were down-regulated at different degrees in the two varieties during cold stress (Fig 3B, S2 Dataset). Meanwhile, serine

acetyltransferase 1 (*SAT1*) (Bra016548) and serine acetyltransferase 3 (*SAT3*) (Bra016589) were both down-regulated under cold stress compared to their control samples (Fig 3B, S2 Dataset). These two enzymes (*SAT*, EC:2.3.1.30) caused L-serine to convert to *O*-acetyl-L-serine (Fig 5). The above two kinds of enzymes played a role in the decomposition of glutamate and serine in biochemical reactions. The down-regulated expression level of the above six genes may be why the content of glutamate and serine increased.

Another enzyme was identified to catalyze S-adenosyl-L-methionine into *S*-adenosyl 3-propylamine, named S-adenosylmethionine decarboxylase proenzyme 1 (*AMD1*, EC:4.1.1.50) (Bra013148) with an up-regulation tendency in response to cold stress in two samples (Figs 3B and 5, S2 Dataset). Methionine content was synchronously determined to decrease in the two varieties during cold stress.

Three transcripts, Bra005012, Bra017051 and Bra007179, annotated in delta-1-pyrroline-5-carboxylate synthase (*P5CS*, EC:2.7.2.11, 1.2.1.41), and transcript Bra017570 annotated in δ-ornithine aminotransferase (*δ-OAT*, EC:2.6.1.13) were identified in our study. The delta-1-pyrroline-5-carboxylate synthase (*P5CS*) and δ-ornithine aminotransferase (*δ-OAT*) are the key enzymes in proline synthesis pathways [55]. The expression level of Bra005012, Bra017051 and Bra007179 showed a increased trend after cold treatment in both plants (Fig 3B, S2 Dataset). However, more *P5CS* expression in L7 than in T2 under cold stress, the log2FC of Bra005012, Bra017051 and Bra007179 in L7 was 2.27, 5.05 and 4.88, while in T2, the log2FC was 0.86, 4.34 and 0.53, respectively. We also found that the expression level of Bra017570 was increased in L7 and decreased in T2 (Fig 3B, S2 Dataset). According to the expression levels of these two enzymes, we speculated that the proline content in the experiment showed an opposite trend in the two plants after cold treatment was related to the two enzymes.

The alanine-glyoxylate transaminase (*AGXT*, EC:2.6.1.44) is catalyze the reaction of glyoxylate into glycine (Fig 5). Two transcripts, Bra037207 and Bra013148, encoding alanine-glyoxylate transaminase were identified, and the expression level of Bra037207 and Bra013148 was upregulated after cold stress in two cultivars. The difference is that the more expression of *AGXT* in L7 than in T2 (Fig 3B, S2 Dataset). The branched-chain-amino-acid transaminase (*ilvE*, EC:2.6.1.42, Bra018463) is catalyzed the conversion between L-Valine and 3-Methyl-2-oxobutanoic acid with downregulated in L7 while upregulated in T2 after cold stress, and the expression of this transcript was higher in T2 than in T7 both in control group and in treatment group (Figs 3B and 5, S2 Dataset). The alpha-aminoadipic semialdehyde synthase (*AASS*, EC:1.5.1.8, 1.5.1.9) is catalyze the L-Lysine into Saccharopine and then formed acetoacetyl-CoA (Fig 5). Tyrosine aminotransferase (*TAT*, EC 2.6.1.5) catalyzes the reversible transamination from tyrosine to form 4-hydroxyphenylpyruvic acid (pHPP), an initial step of the tyrosine conversion [56]. One transcript, Bra037020, annotated in *AASS* and two transcripts (Bra0226866, Bra039489) annotated in tyrosine aminotransferase (*TAT)* were found in our study. All these transcripts were exhibited upregulated tendency in two cultivars after cold stress, and more expression in T2 than in L7 (Fig 3B, S2 Dataset). In addition, we also found that more arginine decarboxylase (Bra011554, EC:4.1.1.19) existed in T2 than in L7 (Fig 3B, S2 Dataset). We suspected that the various expression level of the above genes may be why the various content of glycine, arginine, proline, tyrosine, valine, and lysine in two cultivars after cold stress.

Bra001090, Bra001131, Bra001195, and Bra001289 were simultaneously annotated in alanine, aspartate, and glutamate metabolism (ko00250), arginine and proline metabolism (ko00330), glyoxylate and decarboxylate metabolism (ko00630), nitrogen metabolism (ko00910), and biosynthesis of amino acids (ko01230). Bra016548 and Bra016589 were identified in four KEGG pathways, such as cysteine and methionine metabolism (ko00270), sulfur metabolism (ko00920), carbon metabolism (ko01200), and biosynthesis of amino acids (ko01230). Bra013148 was only detected in cysteine and methionine metabolism (ko00270)

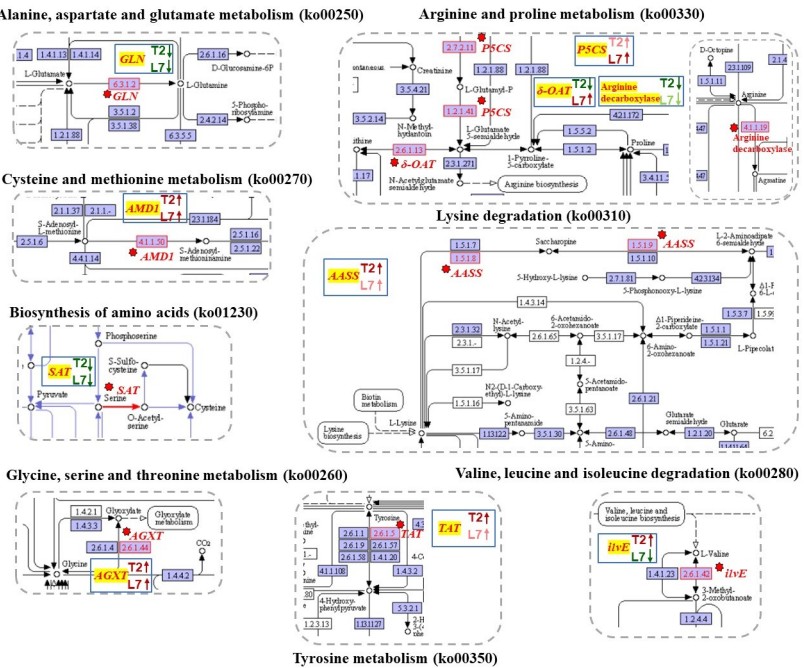

**Fig 5. KEGG pathway of DEGs related in amino acid metabolism.** The images showed that the genes involved in fatty acid synthesis were mapped to the reference pathway using KEGG. The genes annotated in KEGG map were highlighted with red texts and red arrows. The blue box in each pathway showed the expression tendency of genes in two cultivars. The red on the blue box shows upregulated, the green indicates downregulated.

and arginine and proline metabolism (ko00330). Therefore, amino acid metabolism was a very complicated process, and the phenomenon that one gene participates in multiple metabolic pathways was commonly observed.

## Discussion

In northwestern China, chilling damage is a serious problem that limits the yield of many crops [57]. In recent years, some researchers have aimed to introduce winter rapeseed into northwestern China [11]. With breakthroughs in breeding of cold-resistant winter rapeseed, some varieties with strong cold resistance were developed that can be safely planted in most northern regions of China. Now, winter rapeseed has become a new overwintering crop in northern areas, and has brought significant economic and ecological benefits [11, 58].

Low temperature resistance is a complex trait involved in many metabolic pathways and cell compartments in plants [59]. Gene expression, bio-membrane lipid composition, and small molecule accumulation are all influenced by cold stress [60]. To survival under low temperature, more energy carriers have to generate lipids, amino acids, membrane components, and other molecules to improve cell membrane fluidity and structural rearrangement [61, 62]. At low temperature, greater membrane lipid unsaturation appears to be crucial for optimum membrane function [2]. It has been reported that genes involved in fatty acid metabolism were up-regulated under cold stress [63]. The up-regulation of lipid synthesis can strengthen the ability to survive under low temperature. Plants subjected to stress have shown accumulation of amino acids, which may affect the synthesis and activity of some enzymes, gene expression, and redox-homeostasis [64]. When cells suffer with cold temperatures, the changes could be detected in protein and lipid membrane composition, which help the organism to restore metabolite homeostasis, and is considered to be a coping mechanism in this condition [47,

65]. Free fatty acid and amino acids as important components of lipids and proteins were logically selected to be the objects of our study. Cold-resistant plants have been reported to have a higher proportion of unsaturated fatty acids and hence a lower transition temperature for adapting to low temperature [66]. In addition, the several studies have reported the change of the plant pattern of amino acids under cold stress [67]. In plants subjected to cold treatment, to enhance adaptability to low temperature, more energy carriers (e.g., ATP) are consumed to produce lipids, amino acid, and other molecules to further promote cell membrane fluidity and structural rearrangement [68, 69].

In our study, we focused on two types of physiological index difference of FFA and amino acid content in the two varieties between normal and freezing temperature. According to the results, the FFA and vast majority amino acids were significantly increased under cold stress in L7, which is the cold resistant variety. Our study also showed that the glutamate, serine, and alanine were significantly increased after cold treatment, and the threonine, methionine, and leucine decreased with cold treatment in the two varieties. Khaled also found that there were significant increases in the contents of Asp, along with a decrease in Iso/Leu concentrations in *Pinus halepensis* under cold stress [70].

The plasma membrane performs crucial roles in response to low temperature stress, and lipids are the major component of the plasma membrane [71]. In our study, genes involved in fatty acid metabolism, such as *FAD3*, *ADS2*, *DGK*, and *KCS2* were differentially expressed in the two varieties and more highly expressed in L7. *FAD3* and *ADS2* were also called desaturase genes (DES), they increased fluidity of membranes for adaption to cold stress by enhancing their expression [72]. Thus, we speculate these genes with higher expression level contributed to resist the cold stress in L7.

The genes involved in amino acid metabolism, such as *GLN* and *SAT*, were down-regulated in both cultivars under cold stress. There was a negative relation between expression level of these two enzymes and glutamate and serine content, and they were responsible for glutamate and serine regulation. We also found that *AMD1* was up-regulated in two cultivars under cold and made content of methionine decline. The content of proline in plants is often used as a physiological index to judge the resilience of crops [73]. The content of proline in L7 was increased while in T2 decreased. As a speed limiting enzyme in the process of proline synthesis, the expression of gene *P5CS* also reflects the ability of plants to resist stress at the level of molecular expression [74]. The analysis of gene expression in our research reveals that the three genes encoding delta-1-pyrroline-5-carboxylate synthase (*P5CS*) were identified and up-regulated in L7 and T2. However, more *P5CS* was expression in L7 than T2, and the genes encoding *P5CS* was significantly differential expression in L7 but not significantly in T2. In addition, we identified the one gene encoding δ-ornithine aminotransferase (*δ-OAT*) were up-regulated in L7 and down-regulated in T2 after cold treatment. We hypothesis that the reason of various content of proline accumulation in two cultivars is related to the expression of *P5CS* and *δ-OAT*. Moreover, the DEGs involved in amino acid metabolism, such as alanine-glyoxylate transaminase (*AGXT*), branched-chain-amino-acid transaminase (*ilvE*), alpha-aminoadipic semialdehyde synthase (*AASS*), tyrosine aminotransferase (TAT) and arginine decarboxylase were suspected responsible for various content of glycine, arginine, proline, tyrosine, valine, and lysine in two cultivars after cold stress.

In short, our study determined the physiological traits of two winter turnip rape varieties, and a transcriptomic analysis was conducted. The aim was to reveal the molecular mechanisms involved in response to cold stress. A number of DEGs were identified as being involved in fatty acid synthesis and amino acid metabolism. Taken together, these results may facilitate the understanding of the molecular mechanisms related to cold resistance between two varieties of winter turnip rape.

## Conclusions

L7 and T2 are two closely related winter turnip rape cultivars, but their tolerance to cold is quite different. Overall, physiological traits combined with transcriptome sequencing technology were utilized to decipher key genes associated with cold response. Both the physiological traits and transcriptomes indicated higher responsiveness to cold stress in L7, probably because of those genes and metabolites mentioned above. The results of this study provide to a deeper understanding of the highly complex regulatory mechanisms in plants during cold treatment.

## Supporting information

**S1 Fig. The correlations between every two samples among their biological replicates.** (DOCX)

**S2 Fig. qRT-PCR validation and correlation analysis of 19 selected genes between the control and cold treatment of two winter turnip rape varieties.** Grey bars indicate the transcript abundance change based on the FPKM values, according to RNA-seq (left y-axis). Blue lines with standard errors represent the relative expression level, determined by qRT-PCR (right y-axis). The last graph is correlation analysis based on qRT-PCR and RNA-seq data, Pearson's correlation coefficient is 0.9248 ($P < 0.05$). (DOCX)

**S3 Fig. Hierarchical cluster map of all differentially expressed genes in two varieties.** (DOCX)

**S4 Fig. Top 20 enriched pathways for cultivar specific differential expressed genes in L7 and T2.** A. Top 20 enriched pathways for L7 specific cold responsive up-regulated genes; B. Top 20 enriched pathways for L7 specific cold responsive down-regulated genes; C. Top 20 enriched pathways for T2 specific cold responsive up-regulated genes; D. Top 20 enriched pathways for T2 specific cold responsive down-regulated genes. (DOCX)

**S1 Table. L7 and T2 cold-resistance capability in ten regions of Gansu province in China.** (DOCX)

**S2 Table. Dilute original density standard.** (DOCX)

**S3 Table. Mobile phase gradient elution procedure.** (DOCX)

**S4 Table. The primers of 19 transcripts and referenced gene for qRT-PCR validation.** (DOCX)

**S5 Table. The annotation information of 19 transcripts for qRT-PCR.** (DOCX)

**S1 Dataset.** (XLSX)

**S2 Dataset.** (XLSX)

## Acknowledgments

We thank all experts for their insightful comments and expertise that greatly assisted the research. We are also grateful to the Gansu Provincial Key Laboratory of Aridland Crop Science, Gansu Agricultural University and Crop research institute that gave us the experiment condition and access to the samples and thus made the project possible. Eventually, we would like to thank the anonymous reviewers for their comments, criticism and recommendations, which enabled a significant improvement of the manuscript.

## Author Contributions

**Conceptualization:** Yan Fang, Wancang Sun.

**Data curation:** Yan Fang, Lijun Liu, Xuecai Li, Li Ma, Yuanyuan Pu, Jiaojiao Jin, Yuhong Zhao, Yaozhao Xu.

**Formal analysis:** Yan Fang, Junyan Wu, Yun Dong, Bolin Sun, Zaoxia Niu, Wenbo Mi.

**Investigation:** Yan Fang.

**Project administration:** Wancang Sun.

**Writing – original draft:** Yan Fang.

**Writing – review & editing:** Yan Fang, Jeffrey A. Coulter, Junyan Wu, Yun Dong, Wancang Sun.

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
