## [Decision Letter · Decision Letter 0]

23 Nov 2020

PONE-D-20-16938

Identification of differentially expressed genes involved in amino acid and lipid accumulation of winter turnip rape (Brassica rapa L.) in response to cold stress

PLOS ONE

Dear Dr. Sun,

Thank you for submitting your manuscript to PLOS ONE. After careful consideration, we feel that it has merit but does not fully meet PLOS ONE’s publication criteria as it currently stands. Therefore, we invite you to submit a revised version of the manuscript that addresses the points raised during the review process.

We look forward to receiving your revised manuscript.

Kind regards,

Kun Lu, Ph.D.

Academic Editor

PLOS ONE

Journal Requirements:

Reviewers' comments:

Reviewer's Responses to Questions

**Comments to the Author**

1. Is the manuscript technically sound, and do the data support the conclusions?

Reviewer #1: Partly

Reviewer #2: Yes

2. Has the statistical analysis been performed appropriately and rigorously? 

Reviewer #1: Yes

Reviewer #2: Yes

3. Have the authors made all data underlying the findings in their manuscript fully available?

Reviewer #1: No

Reviewer #2: Yes

4. Is the manuscript presented in an intelligible fashion and written in standard English?

Reviewer #1: No

Reviewer #2: Yes

5. Review Comments to the Author

Reviewer #1: 1. Why the author choose seedling stage sample to interpret the lipid accumulation. I think the cold tolerance have more relation with the sugar metabolism than the lipid metabolism.

2. line 217-218, rewrite it and the writing needs to be polished for the whole text.

3. In figure 1，the phenotype is not obvious under the cold stress, and the soil looks different in the pod, it would be better show readers the root phenotype since the whole research is on root..

4. The figures quality are poor.

5. Since there are 9 amino acids were different between L7 and T2, which include Proline, and please use it as a example by using the transcriptome data to interpret this difference.

6. line 365, correct 4°;

7. How about the DEG involved in phytohormone synthesis or signaling, please add this analysis to the MS;

8. Table1 move to the supplimentary file;

9. Correct brassica rapa in italic;

Reviewer #2: The manuscript deals with the analysis of genes related in response to cold stress in Brassica rapa L. The study is approached and contain intersting results, however, the manuscript needs revisions before accepting for publication. Following main points need due consideration.

1. P3-line 105-106, “until the plants grew to the six leaf stage, then the seedlings were treated with different conditions including control (CK) and cold stress”, various crop varieties have different growth periods, “grew to the six leaf stage”, the time statement can be more clearer, such as “three-week-old”, or “30 day old” and so on.

2. In materials and methods, the author should indicate photosynthetic photon flux density and relative humidity, it has been reported that light intensity affects the tolerance of plants to low temperature.

3. In Figure 1, the authors may consider the possibility of statistical analysis of phenotypic results, such as total leaf area, leaves fresh weight, root fresh weight, leaves dry weight, or root dry weight.

4. In Figure 3, transcript id should be added the annotation of gene name, such as “FAD3 (Bra××××××)”, and the author's analysis of Figure 3 is deficient, should make further analysis.

5. In transcriptome results, GO_enrichment and the KEGG map of amino acid metabolism and fatty acid synthesis should be added to the article or supporting information, it is helpful for readers to analyze the function of key genes.

6. Through out the manuscript lack of possible raising mechanisms, authors are encouraged to include some recent reports discussing the signalling regulation mechanisms.

6. PLOS authors have the option to publish the peer review history of their article (what does this mean?). If published, this will include your full peer review and any attached files.

Reviewer #1: No

Reviewer #2: No

---

## [Author Response · Author response to Decision Letter 0]

18 Dec 2020

Response to reviewers

Reviewer #1: 

1. Why the author choose seedling stage sample to interpret the lipid accumulation. I think the cold tolerance have more relation with the sugar metabolism than the lipid metabolism.

Response: Thank you for your comments. There two reason why we choose seedling stage sample to interpret the lipid accumulation. At first, in cold and arid areas of northern China, the winter rapeseed was sown from later August to early September. Generally, seedlings grew to the fourth or fifth leaf stage, which experienced a significant cooling environment. When the seedings grew to the six-leaf stage, entered the overwintering stage. The cold resistance of the plants in the six-leaf stage determined whether the winter rapeseed could survive in the winter safely. Secondly, the plants show complex adaptations to freezing that prevent cell damage caused by cellular dehydration. Lipid remodeling of cell membranes during dehydration is one critical mechanism countering loss of membrane integrity and cell death. Therefore, the seedling samples were selected for sequencing. In addition, as you said, the cold resistance is strongly related to sugar metabolism in plant. However, the focus of our work is on amino acid and lipid accumulation, which is very different from other work, and this is also the major contribution of our work. 

2. line 217-218, rewrite it and the writing needs to be polished for the whole text.

Response: According to the comment of the reviewer, we have made correction in our revised paper (line 216-217).

3. In figure 1, the phenotype is not obvious under the cold stress, and the soil looks different in the pod, it would be better show readers the root phenotype since the whole research is on root.

Response: We are sorry for our negligence. We checked it and changed the figure in our revised paper (Figure 1). Moreover, we added the root phenotype in revised paper.

4. The figures quality are poor.

Response: We are sorry for our negligence. We did our best effort to modify the characters of figures in our revised paper.

5. Since there are 9 amino acids were different between L7 and T2, which include Proline, and please use it as a example by using the transcriptome data to interpret this difference.

Response: Thank you for your comments. According to your suggestion, we conducted this analysis, and added this part in our revised paper (line 390-425).

6. line 365, correct 4°;

Response: We are sorry for our negligence, we checked it and made correction (line 364 in revised paper).

7. How about the DEG involved in phytohormone synthesis or signaling, please add this analysis to the MS;

Response: Thank you for pointing this out. In our previous study, we made analysis of DEG involved in phytohormone synthesis and signaling. In this work, we focus on the relation between amino acid, lipid accumulation and cold stress in winter rapeseed. We still appreciate your advice.

8. Table1 move to the supplimentary file;

Response: Thank you for your comments. Table 1 reflected the whole RNA-seq characteristic in each sample. We think that it is important to further study in this area. So, we were insisted on putting table 1 in the body of text.

9. Correct brassica rapa in italic;

Response: We are sorry for our negligence, we checked whole text and made correction.

Reviewer #2: The manuscript deals with the analysis of genes related in response to cold stress in Brassica rapa L. The study is approached and contain intersting results, however, the manuscript needs revisions before accepting for publication. Following main points need due consideration.

1. P3-line 105-106, “until the plants grew to the six leaf stage, then the seedlings were treated with different conditions including control (CK) and cold stress”, various crop varieties have different growth periods, “grew to the six leaf stage”, the time statement can be more clearer, such as “three-week-old”, or “30 day old” and so on.

Response: We are sorry for our negligence. In cold and arid areas of northern China, the winter rapeseed was sown from later August to early September. Generally, seedlings grew to the fourth or fifth leaf stage, which experienced a significant cooling environment. When the seedings grew to the six-leaf stage (60 days after sown), entered the overwintering stage (probably October to November). So, we're used to writing “six-leaf stage”. According to your advices, we were rewrite it within 60 days in revised paper (line 104).

2. In materials and methods, the author should indicate photosynthetic photon flux density and relative humidity, it has been reported that light intensity affects the tolerance of plants to low temperature.

Response: We are sorry for our negligence. We added the photosynthetic photon flux density and relative humidity in our revised paper (line 108-109). 

3. In Figure 1, the authors may consider the possibility of statistical analysis of phenotypic results, such as total leaf area, leaves fresh weight, root fresh weight, leaves dry weight, or root dry weight.

Response: Thank you for your advices and we appreciated it. In our previous study, the fresh matter, dry matter, root/shoot ratio and main root diameter of L7 and T2 under cold treatment were described in detail [Morphology and physiological characteristics of cultivars with different levels of cold-resistance in winter rapeseed (Brassica campestris L.) during cold acclimation. Sci Agric Sin 2013, 22]. In this study, the cold treatment was decreased from 22℃ to -4℃ at a cooling rate of 2℃ per hour. After treated at low temperature, we selected the leaves firstly showed wilting symptoms as time point of sampling. So, in figure 1 (old), there are no visible significant different between L7 and T2. This caused a misunderstanding to the reader. In our revised paper, we replaced the figure 1 (old) with new picture and added the root morphology of two cultivars (revised paper Figure 1A and 1B).

4. In Figure 3, transcript id should be added the annotation of gene name, such as “FAD3 (Bra××××××)”, and the author's analysis of Figure 3 is deficient, should make further analysis.

Response: We are very sorry for the misunderstanding caused by our incorrect expression. The 19 genes used for validated were randomly selected from RNA-seq data, were not selected from differentially expressed genes (DEGs). In our paper, we were mentioned “Nineteen genes were selected for qRT-PCR validation to verify the RNA-seq data, involving genes in the amino acid, sugar, lipid, and plant hormone signal metabolic pathways which have already been reported to be related with cold stress.”, it is because of the annotation information of these 19 genes suggested that these genes are annotated into these metabolic pathways. Moreover, we found that the genes of Bra011511, Bra004136, Bra008792, Bra007142 and Bra036828 respectively annotated in aldehyde dehydrogenase, glutamate decarboxylase 2, chalcone synthase, chalcone-flavanone isomerase 1 protein, flavanone 3-hydroxylase 1 protein were related cold stress by looking at the references. Since the purpose of the qRT-PCR experiment is verify the accuracy of sequencing data, and our results also show that our sequencing data are reliable. We think that this analysis is also feasible for this part. As for your suggestions, we also added the annotation information of the 19 transcripts in revised paper, which is shown in Table S5.

5. In transcriptome results, GO_enrichment and the KEGG map of amino acid metabolism and fatty acid synthesis should be added to the article or supporting information, it is helpful for readers to analyze the function of key genes.

Response: Thank you for your comments. According to your suggestion, we conducted this analysis, and added this part (Figure 4 and 5) in our revised paper.

6. Through out the manuscript lack of possible raising mechanisms, authors are encouraged to include some recent reports discussing the signalling regulation mechanisms.

Response: Thank you for pointing this out. In our previous study, we made analysis of DEG involved in signaling regulation [Transcriptome Analysis Reveals Key Cold-Stress-Responsive Genes in Winter Rapeseed (Brassica rapa L.). Int J Mol Sci 2019, 20, doi:10.3390/ijms20051071]. In this work, we focus on the relation between amino acid, lipid accumulation and cold stress in winter rapeseed. We still appreciate your advice.

---

## [Decision Letter · Decision Letter 1]

4 Jan 2021

Identification of differentially expressed genes involved in amino acid and lipid accumulation of winter turnip rape (Brassica rapa L.) in response to cold stress

PONE-D-20-16938R1

Dear Dr. Sun,

We’re pleased to inform you that your manuscript has been judged scientifically suitable for publication and will be formally accepted for publication once it meets all outstanding technical requirements.

Kind regards,

Kun Lu, Ph.D.

Academic Editor

PLOS ONE

Additional Editor Comments (optional):

Reviewers' comments:

Reviewer's Responses to Questions

**Comments to the Author**

1. If the authors have adequately addressed your comments raised in a previous round of review and you feel that this manuscript is now acceptable for publication, you may indicate that here to bypass the “Comments to the Author” section, enter your conflict of interest statement in the “Confidential to Editor” section, and submit your "Accept" recommendation.

Reviewer #1: All comments have been addressed

Reviewer #2: All comments have been addressed

2. Is the manuscript technically sound, and do the data support the conclusions?

Reviewer #1: Yes

Reviewer #2: Yes

3. Has the statistical analysis been performed appropriately and rigorously? 

Reviewer #1: Yes

Reviewer #2: Yes

4. Have the authors made all data underlying the findings in their manuscript fully available?

Reviewer #1: Yes

Reviewer #2: Yes

5. Is the manuscript presented in an intelligible fashion and written in standard English?

Reviewer #1: Yes

Reviewer #2: Yes

6. Review Comments to the Author

Reviewer #1: (No Response)

Reviewer #2: (No Response)

7. PLOS authors have the option to publish the peer review history of their article (what does this mean?). If published, this will include your full peer review and any attached files.

Reviewer #1: No

Reviewer #2: No

---

## [Editor Report · Acceptance letter]

21 Jan 2021

PONE-D-20-16938R1 

Identification of differentially expressed genes involved in amino acid and lipid accumulation of winter turnip rape (*Brassica rapa* L.) in response to cold stress 

Dear Dr. Sun:

I'm pleased to inform you that your manuscript has been deemed suitable for publication in PLOS ONE. Congratulations! Your manuscript is now with our production department. 

Kind regards, 

on behalf of

Dr. Kun Lu 

Academic Editor

PLOS ONE